# Egg Sterilisation of Irradiated *Nezara viridula* (Hemiptera: Pentatomidae)

**DOI:** 10.3390/insects11090564

**Published:** 2020-08-24

**Authors:** Kiran Jonathan Horrocks, Taylor Welsh, Jim E Carpenter, David Maxwell Suckling

**Affiliations:** 1School of Biological Sciences, University of Auckland, Private Bag 92019, Auckland Mail Centre, Auckland 1142, New Zealand; Max.Suckling@plantandfood.co.nz; 2The New Zealand Institute for Plant and Food Research Limited, Private Bag 4704, Christchurch Mail Centre, Christchurch 8140, New Zealand; Taylor.Welsh@plantandfood.co.nz; 3Better Border Biosecurity, Lincoln 7608, New Zealand; 4Retired from Crop Protection and Management Research Unit, USDA-ARS, Tifton, GA 31793, USA; jecarpent@gmail.com

**Keywords:** sterile insect technique, sterile males, southern green stink bug, green vegetable bug, Pentatomidae, irradiation, eradication

## Abstract

**Simple Summary:**

Certain stink bugs are emerging as a serious threat to food production globally, as they invade new areas and feed on a wide range of crops. The control of these pests relies primarily on potentially harmful pesticides, but the increasing threat posed by stink bug pests has sparked investigation into environmentally-friendly methods that do not exert serious impacts on other species. One method receiving attention is the Sterile Insect Technique (SIT), which involves sterilising large numbers of a pest, through radiation exposure, and releasing them into the wild pest population where mating results in unfertilised eggs that do not hatch. In support of recent studies on SIT for the brown marmorated stink bug, we aimed to ascertain the feasibility of the method for another stink bug pest, the green vegetable bug. We exposed the insects to increasing levels of radiation and allowed mating to occur. Virtually all of the resulting eggs were sterile and did not hatch at the higher radiation doses tested. These results could be used to inform the potential development of SIT against stink bug pests, and in some circumstances, could form the basis of potential eradication programmes against new invasions.

**Abstract:**

*Nezara viridula* Linnaeus (Hemiptera: Pentatomidae) is a polyphagous pest of a wide range of economically important crops. Because the control of this species and other pentatomids relies primarily on insecticide application, investigation into the Sterile Insect Technique (SIT) is warranted. We aimed to investigate the irradiation biology of *N. viridula* for the potential application of SIT against this pest. Male and female *N. viridula* were gamma-irradiated at doses between 4 and 28 Gy and mated with both irradiated and nonirradiated conspecifics. Sterility of the resulting eggs followed a dose-response in each case. Irradiated males crossed with untreated females showed higher F_1_ egg sterility than crosses where the female was irradiated. The greatest F_1_ egg sterility was observed when both parents were irradiated. There was no obvious dose-response for the longevity of irradiated males, and for the fecundity of nonirradiated females mated with irradiated males. The fecundity of irradiated females appeared to decrease with irradiation dose. These results can be applied to a potential future application of SIT against *N. viridula*, but predominantly supports the ongoing development of SIT for *Halyomorpha halys* Stål (Hemiptera: Pentatomidae) and hemipteran pests in general.

## 1. Introduction

*Nezara viridula* Linneaus (Hemiptera: Pentatomidae) is a highly polyphagous cosmopolitan pest of a wide range of economically important horticultural crops in its invaded ranges of the Americas, Asia, Australasia, and Europe [1,2,3]. The pentatomid family is known for other invasive economic pests, most recently *Bagrada hilaris* Burmeister (bagrada bug) (Hemiptera: Pentatomidae) and *Halyomorpha halys* Stål (brown marmorated stink bug) (Hemiptera: Pentatomidae) [4]. Pentatomids feed preferentially on seeds or immature fruits by inserting their stylet, extruding saliva, and sucking up the dissolved contents. The resulting damage can include discolouration around the puncture sight and chalky air spaces where cell contents were removed, causing spoilage [5]. Control of *N. viridula* and other pentatomids in most invaded regions relies heavily on chemical insecticides, including organophosphates [6,7,8]. Their significant pest status warrants investigation into more environmentally-friendly and target specific methods for pentatomid pests, including biological control and mating disruption through vibrational signalling [8,9,10,11].

One organic strategy is the Sterile Insect Technique (SIT), which is species-specific without nontarget risks or other in-field environmental concerns [12], and is capable of significantly contributing to eradication of mainly dipteran and lepidopteran pests [13]. SIT generally involves sterilising mass-reared males of the target pest via gamma or X-ray radiation exposure and their subsequent release into the target pest population. The wild females that mate with sterilised males lay unfertilised eggs, resulting in a reduction in numbers, or in a few cases inherited sterility, in the F_1_ generation [13,14]. Investigation into the irradiation biology of Hemiptera for SIT is scant, with Welsh et al. [11] recently ascertaining the irradiation biology of male *H. halys*. Studies that have irradiated *N. viridula* have focused primarily on how life-stage effects radiation tolerance [15], and how substerilising doses impact reproductive fitness [16,17]. Other Hemiptera species for which irradiation dosimetry has been calculated for potential application of SIT include *Diaphorina citri* (Hemiptera: Liviidae) [18], *Eurygaster maura* Linnaeus (Hemiptera: Scutelleridae), *Eurygaster austriaca* Schrank (Hemiptera: Scutelleridae) (sun pest) [19,20], and species belonging to the Aleyrodidae (whiteflies), Diaspididae (scale insects), Cicadellidae (leafhoppers), Coccidae (scale insects), and Pyrrhocoridae (red bugs) families [19].

Current investigations into tools for contributing to the eradication of *H. halys* have led to the proposal for live trapping pre-overwintering bugs [21], irradiation and shipping to another jurisdiction for use in an eradication [22], as well as sterilising a classical biological control agent so that the contribution to eradication is followed with its own extinction [23]. Application of SIT during an eradication overcomes some of the concerns of direct fruit damage from released sterile insects, since the area and duration would be limited and a successful outcome might be achievable under certain circumstances. However, given that pentatomid SIT development is in its infancy, many crucial factors must be investigated before confirming the feasibility of implementation [11,24,25].

In support of the increasing body of literature surrounding pentatomid SIT, and a recent study targeting new tactics against *H. halys* [11], we aimed to ascertain the dosimetry for the irradiation biology of adult male and female *N. viridula*. Because postirradiation retention of fitness in sterile insects is essential for the success of SIT programmes [25], pentatomid species were also investigated for this model.

## 2. Materials and Methods

Insect rearing: The insects were reared at the United States Department of Agriculture (USDA) Agricultural Research Service, Tifton, GA (ambient 26 °C, 16:8 L:D, 87% R.H.). The nymphs were reared in a 450 mL plastic cup with a filter paper liner, a water source, green beans, and sunflower seeds. Green beans in each cage were refreshed three times weekly. Sunflower seeds were changed approximately weekly. The water source was refreshed daily.

Irradiation: Male and female *N. viridula* were irradiated on the day of fifth instar emergence. Those in the control (0 Gy) treatment were exposed to the same environmental conditions as the irradiated insects. Gamma radiation was applied using a ^60^Co source (Gammacell 220 irradiator, Nordion, Ottawa, Canada) at doses of 4, 8, 12, 16, 20, 24, and 28 Gy. The dose rate was 3.3 Gy/min. This was also repeated for *N. viridula* adults 24 h after emergence from the fifth instar, but a substantially lower fecundity resulted in a paucity of data that were less suitable for analysis.

Experiments: For the first experiment, 15 fifth instar males were irradiated per dose, which were later mated to 15 untreated virgin adult females. This led to the evaluation of hatch for 6–60 egg batches per irradiation dose (mean 34.37). The second experiment irradiated 15 fifth instar females per dose, which were then mated with untreated virgin adult males. This led to the evaluation of hatch for 1–44 egg batches per irradiation dose (mean 21.36). The final experiment irradiated 15 fifth instar pairs per dose, which led to the evaluation of hatch for 7–41 egg batches per irradiation dose (mean 21.43).

Crosses and assessments: The aforementioned crosses of irradiated and nonirradiated virgin *N. viridula* with conspecifics of the opposite sex were mostly established within 24 h of emergence as an adult for both sexes. Otherwise, individual adults were maintained in labelled 450 mL cups and provided with food and water until they were crossed no later than five days after adult emergence. Crosses were carried out within a 450 mL cup containing an approximately 2.5 cm long section of green bean, two or three sunflower seeds, and a water source, with resulting eggs being removed and counted. Each resulting F_1_ egg mass was placed into a 6 cm Petri dish and labelled with the corresponding dose and code number of its parents, and date of collection, with the number of full and empty eggs within masses being counted. Upon the emergence of first instar larvae, the F_1_ egg masses were placed into a 9 cm petri dish with a green bean for food. When the larvae abandoned the egg mass, the number of eggs that failed to develop or emerge were counted to assess the effect of radiation treatment on the degree of sterility. Any nymphs that reached the third instar were then transferred to a 450 mL cup and additionally given sunflower seeds and water. The number of successfully emerged F_1_ insects surviving to each instar was recorded for each egg mass, in addition to the time taken for each offspring to become an adult. Irradiated insects in all crosses were reared until death to examine the potential impact of each radiation treatment on longevity and were checked each day with any mortality recorded.

Statistical analyses: The egg sterility data were subject to analysis of variance (ANOVA), without correction for the control. Due to a lower residual variance considering curve fitting, a logarithmic dose-response was followed. The fecundity and longevity data were also subject to ANOVA.

## 3. Results

Irradiated males × untreated females. For irradiated male × untreated female crosses, egg sterility followed a dose-response, with the mean percentage egg sterility being significantly impacted by dose (F = 26.43, *p* < 0.0001) (Figure 1). The scatter in the data reveals high variability around the curve fitting, though 98.14% sterility was achieved by irradiating males with 16 Gy. This was a significant increase from 69.06% at 12 Gy. Egg sterility reached 100% at 20 and 24 Gy but fell slightly to 99.54% at 28 Gy.

Irradiated females × untreated males. A similar dose–response pattern was observed for irradiated female × untreated male crosses (F = 10.27, *p* < 0.0001) (Figure 2). A dose of 28 Gy induced 100% egg sterility. However, sterility was lower than anticipated for females irradiated at 24 Gy, which resulted in 60.47% egg sterility. This is lower than the egg sterility observed in the 8, 12, and 16 Gy treatments.

Irradiated males × irradiated females. Percentage egg sterility was also significantly impacted by dose for the bi-parental irradiation crosses (F = 21.82, *p* < 0.0001) (Figure 3). Irradiating both parents with 16, 20, and 24 Gy resulted in 100% egg sterility, with 28 Gy inducing 99.24% sterility. Higher doses are therefore required to induce complete sterility when irradiating only one of the parents, regardless of sex.

Fecundity: Overall, there was no discernible relationship between paternal irradiation dose and the mean number of eggs laid by females that were mated to those males (F = 1.06, *p* = 0.39) (Figure 4). However, there was a sudden drop in oviposition rates, and only 20 eggs per female were laid at 20 Gy. At 4 and 28 Gy, male irradiation dose resulted in female partners laying a similar number of eggs to the control, but females mated with 12 Gy irradiated males expressed the highest fecundity at 182 eggs per female. The percentage of females ovipositing after mating with irradiated males followed a similar pattern (Figure 4). Conversely, when the female partner was irradiated, there was a statistically significant decrease in the mean number of eggs laid as the irradiation dose they were subjected to increased (F = 2.84, *p* = 0.01) (Figure 5).

Longevity of irradiated adults: For males utilised in the irradiated male × untreated female crosses, there was no discernible relationship between irradiation dose and longevity for the doses tested (F = 1.72, *p* = 0.11) (Figure 6). The mean longevity for nonirradiated male *N. viridula* was 32.6 days. The lowest mean longevity was expressed by 24 Gy irradiated males at 13.75 days, whereas the highest was 16 Gy irradiated males at 49.25 days. Conversely, irradiated females that were crossed with untreated males showed a statistically significant decrease in longevity with increasing irradiation dose (F = 2.96, *p* = 0.008) (Figure 7). Mean longevity for nonirradiated female *N. viridula* was higher than that for nonirradiated males at 52.21 days. The lowest mean longevity was expressed by 28 Gy irradiated females at 17.64 days, which is again higher than the lowest mean longevity for males. The highest mean longevity was shown by the control females.

Due to the high second instar mortality in both the treatments and controls, comparative interpretation of F_1_ nymph mortality due to one or both parents’ exposure to radiation could not be assessed.

## 4. Discussion

### 4.1. Development of SIT for N. viridula and Pentatomid Pests

Aside from the untreated male × irradiated female crosses, >99% sterility was achieved at doses of 16 Gy and above, suggesting that sterility is achieved at a lower dose for irradiated males. However, the dose required to achieve sterility may be higher than this because the unexpectedly sudden increase in sterility from males treated between 12 and 16 Gy may have been caused by variation in received dosimetry due to dose rates varying spatially within a Petri dish. The unexpectedly low egg sterility for irradiated females crossed with untreated males may have also been influenced by a dose field variation. This is a well-known problem when irradiating insects to achieve sterility [25]. To account for this, a minimum acceptable dose that is slightly greater than that required to achieve sterility is generally defined for the target pest species [26]. Considering that 16 Gy resulted in almost complete F_1_ egg sterility for irradiated male × untreated female crosses, these findings are similar to the irradiation biology of *H. halys*, another pentatomid pest irradiated at the same facility [11]. The observation that male sterility was achieved at lower doses than female sterility is common amongst hemipteran families, including the Pentatomidae [15,26], and may be due to immaturity of female oocytes at the time of irradiation, thus avoiding sterilisation [26,27]. This supports the recommendation of Welsh et al. [11], who did not irradiate females, that an SIT programme for *H. halys*, and possibly other Pentatomidae, should involve the release of sterile males. However, in some hemipteran species females require lower doses than males to achieve sterility, such as for *Dysdercus koenigii* Fabricius (Hemiptera: Pyrrhocoridae) (red cotton stainer) [28,29] and *Trialeurodes vaporariorum* Westwood (Hemiptera: Aleyrodidae) (greenhouse whitefly) [30].

The fecundity of females mated with irradiated males showed a similar pattern to that observed in *H. halys* [10], where the mean number of eggs laid by females remained relatively constant at doses up to 28 Gy—the maximum dose tested. Srivastava and Deshpande [28] also found that irradiating *D. koenigii* males did not impact female fecundity. This relationship might therefore be consistent amongst Pentatomidae, suggesting that the mating vigour of sterile males could be retained. This is important because sterile males must compete with wild males to mate with wild females, and successfully transfer sterile sperm and/or accessory fluid for the SIT to be successful [31,32,33]. The unexplained sudden drop in fecundity for females mated with 20 Gy irradiated males suggests a problem with those individuals because it was not supported as a trend. Furthermore, our finding that fecundity decreased with dose when females were irradiated is consistent with studies on SIT for Lepidoptera, which also found no significant change in fecundity when males were irradiated [34,35,36,37]. This further promotes a sterile male release strategy for *N. viridula* and potentially other pentatomid pests, because it is unlikely that sterile females will retain competitive ability amongst wild females within a target population.

The lack of a clear dose-response for the longevity of irradiated male *N. viridula*, as well as the similarity between the control and irradiated groups, again suggests that fitness was maintained in sterile male individuals [31]. However, the negative relationship between longevity and dose for irradiated females in turn indicates a negative fitness impact for females. This further supports the potential for sterile male *N. viridula* to exert competitive and reproductive fitness amongst wild conspecifics [31], and promotes the male release strategy in pentatomid SIT. Average longevity for lab-reared mated *N. viridula* varies widely between studies, with some observing similar results to the control male and female groups in this study [38,39,40].

Whether inherited sterility is attained by any surviving offspring of irradiated parents is worth consideration for the development of an SIT regime [14]. LaChance et al. [41] found some evidence of inherited sterility in the hemipteran *Oncopeltus fasciatus* Dallas (Hemiptera: Lygaeidae) (large milkweed bug). The mechanism for inherited sterility in this species was likely caused by broken chromosome fragments of irradiated parents. This is potentially different from the mechanism for induced inherited sterility in Lepidoptera, which has been successful in controlling pests in this order [42,43]. Stringer et al. [44] assessed the chromosomes undergoing meiosis within the testes of 40 Gy irradiated male *N. viridula*, finding fragmentation and abnormal division of chromosomes, though also suggested that a lower dose is required due to impacts on competitive fitness. It is, therefore, possible that a form of inherited sterility could occur amongst the surviving offspring of irradiated *N. viridula* parents, which Welsh et al. [11] found some evidence for in *H. halys*.

Despite the status of *N. viridula* as a highly cosmopolitan, polyphagous pest of a number of economically important horticultural crops [3], SIT for this species, and for pentatomid pests in general, is not under active development [19]. A lack of an artificial diet to support mass-rearing to support sterile releases may be part of the problem. This study contributes to the potential development of SIT for this species, but predominantly promotes the use of SIT for pentatomid pests, complementing the results found by Welsh et al. [11], who ascertained the irradiation biology for *H. halys*. The similar irradiation biology of *N. viridula* and *H. halys* [11] suggests that irradiation dosimetry may be similar throughout this taxonomic group. The remaining Pentatomoidea that have been irradiated for the potential purpose of SIT are *Eurygaster maura* Linnaeus (Hemiptera: Scutelleridae) and *Eurygaster austriaca* Schrank (Hemiptera: Scutelleridae) (sun pest) [19,20]. However, in this case, 60 Gy produced the best results for male sterility, which is substantially greater than the required dose for male *N. viridula* and *H. halys*, though SIT programmes have yet to be applied to these species [20]. Other hemipteran pest families that have been successfully sterilised also required larger irradiation doses than *N. viridula* and *H. halys* [19]. For example, *D. koenigii*, an important pest of cotton [45], required a dose of at least 40 Gy to induce sufficient male sterility [28,46]. Although this study is useful for the potential implementation of SIT for *N. viridula* and *H. halys*, irradiation dosimetry for different hemipteran families is apparently variable, meaning that pest species belonging to this order, as well as Pentatomidae, will require individual consideration. In fact, recalibration of sterility is essential where more than one irradiator is used [22].

### 4.2. Application of SIT for N. viridula and Other Pentatomid Pests

When exploring SIT options for *N. viridula*, and hemipteran pests in general, it is important to consider how their biology may affect the application of such programmes, and how this could differ from the application of SIT in Diptera and Lepidoptera, for which the technique is well established [37,47,48]. Because the reproductively mature life-stage (adult) must be mass-released for SIT [31], this may lead to an initial increase in damage if targeting Hemiptera because the adult life-stage feeds and causes crop damage [11,31]. However, this could be negated with early season releases so as to not coincide with peak fruiting and to achieve a high overflooding ratio of sterile to wild individuals [49]. This is generally not an issue when mass-releasing sterile adult Diptera or Lepidoptera as the larva is usually the damaging life-stage [50]. Multiple mating by wild female pentatomids [51] is an important consideration for the success of SIT because they could mate with both sterile and wild males [52], and it could also stimulate an increase in fecundity [53]. However, if a suitable overflooding ratio of sterile males is defined and maintained, this should not negatively impact the success of SIT [22,52]. Additionally, the strong dispersal ability of pentatomids such as *H. halys* and *N. viridula* [54,55] could adversely impact the success of SIT through natural immigration rates into the targeted area. A sound consideration of overflooding ratios should again overcome this issue [52].

Furthermore, a species’ biology can affect the feasibility of rearing it in the laboratory by influencing, for instance, quality control, diet regimes, and cost [24]. Rearing techniques are well established for *N. viridula* and *H. halys* [56,57,58], though mass-rearing enough individuals for sterile releases may be costly. It is argued that this would be the primary limitation to the implementation of SIT for *N. viridula*, citing a rearing cost of USD 4000 per 1000 insects [8]. However, *N. viridula* and *H. halys*, as well as other pentatomids, often occur in large aggregations prior to reproductive diapause during the cooler months [59,60,61]. Consequently, large numbers could be collected and stockpiled in cold storage, to induce diapause, until the pest’s feeding season begins or whenever sterile release is required [11,22,24]. In an SIT programme, stockpiling large numbers of *N. viridula* or other pentatomid pests could also involve mass-rearing, especially during winter for seasonal pests, and cold storing the individuals that do not get released immediately [24,62]. Employing these methods could alleviate concerns regarding the cost of mass-rearing pentatomid pests, and therefore, increase the efficacy of a potential SIT programme.

### 4.3. Data Gap—F_2_ Mortality

We were not able to assess the sterility of the few surviving F_1_ offspring from irradiated parents due to the high second instar F_1_ mortality in the control, which prevented comparison. For *H. halys*, a binary response was observed in the surviving F_2_ offspring from irradiated grandfathers, whereby the F_2_ egg masses were either fully sterile or showed no noticeable effect on egg hatch, with no relationship to dose [11]. Following the fate of the F_2_ generation from irradiated grandparents for *N. viridula* would allow a comparison to these findings, and would be useful for implementing a potential SIT programme as it would provide a clearer understanding of risk from any surviving F_1_ offspring [37].

### 4.4. Future Needs

Both mass-rearing and irradiation can potentially affect the quality and competitive fitness of individuals being released as part of an SIT programme. The key variables influencing fitness and quality in mass-reared and irradiated insects are courtship, dispersal, longevity, mating, and sperm transfer, which must stand-up against these factors within the wild target population [25,63]. We considered dosimetry effects on female oviposition, and irradiated male and female longevity for *N. viridula*, though the remaining indicators of fitness and quality, and the protocols required to ensure these [63,64], must still be assessed before the application of an SIT programme. As research is commencing on SIT for hemipteran pests, programmes would also benefit from a research-based consideration of the cost of mass-rearing a given target species, including the facilities required.

The concern of ephemeral crop damage caused by released sterile adults must be investigated when researching the potential of SIT for *N. viridula* and other pentatomid pests, before it is applied [11,31]. Prior assessment will, therefore, need to quantify the likely amount of adult damage that would occur to ensure that this will not outweigh the positive impact of SIT on the amount of feeding from the wild population. Tolerance of crop damage would also likely be higher when SIT is employed for eradication over long-term control.

The sterility of the few surviving offspring from irradiated parents should also be ascertained for the development of an SIT programme for *N. viridula*. This would demonstrate whether any fertile pests will occur in the environment as a result of sterile mass-releases. The challenge of attaining high F_1_ survivability in the control when rearing pentatomids requires attention, with artificial diets not yet as effective as natural diets [57,65].

A rigorous estimate of the ideal overflooding ratio of sterile-wild insects is crucial prior to the implementation of any SIT programme because the chance of a wild insect mating with a sterile insect increases with an increasing number of the latter in relation to the former [52,66]. Overflooding ratios are often estimated through population modelling, based on the percent sterility and biological parameters of the target pest that are relevant to population size over time [52,67]. This would also be pertinent to the development of SIT for *N. viridula* and other pentatomid pests, and could inform the practicality of mass-release at different target pest densities, whereby eradication is more feasible at low densities [68].

## 5. Conclusions

We suggest that irradiating male *N. viridula* would be feasible, and the most efficient method for a potential SIT programme, which aligns with the findings of similar studies using the closely related *H. halys* [11,22]. When males were irradiated, >99% sterility was achieved at doses of 16 Gy and above, whereas when the female partner was irradiated, complete sterility was only achieved at 28 Gy. Unlike females mated with irradiated males, irradiated females experienced a decrease in fecundity as irradiation dose increased. Furthermore, irradiated males showed no relationship between dose and longevity, whereas irradiated females exhibited a decrease in lifespan as the dose increased. This study contributes to the limited research on the scope for SIT against current and emerging pentatomid, and even hemipteran, pests. However, various other factors must first be considered.

## Figures and Tables

**Figure 1 insects-11-00564-f001:**
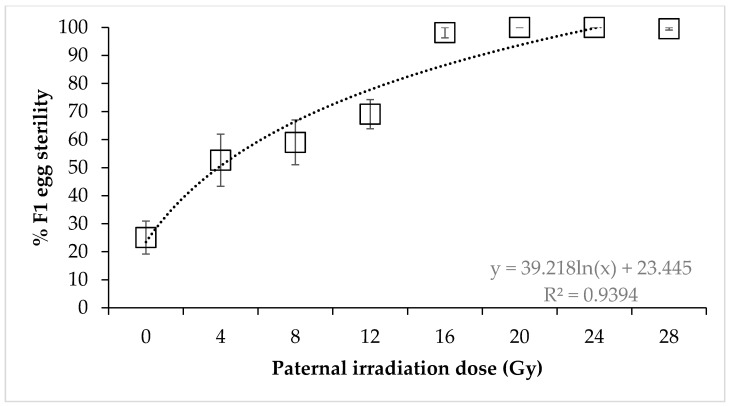
Sterility of *Nezara viridula* eggs after crosses of untreated females with irradiated males. Error bars show standard error (n = 15 crosses per dose).

**Figure 2 insects-11-00564-f002:**
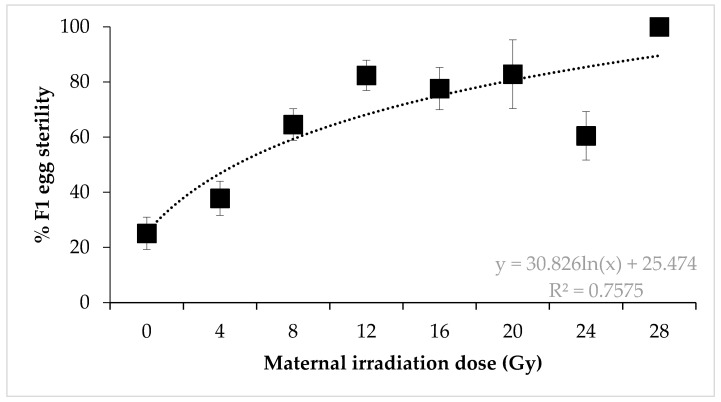
Sterility of *Nezara viridula* eggs after crosses of irradiated females with untreated males. Error bars show standard error (n = 15 crosses per dose).

**Figure 3 insects-11-00564-f003:**
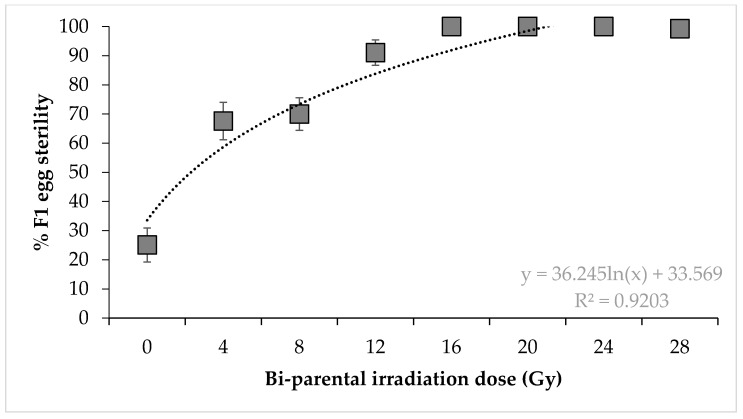
Sterility of *Nezara viridula* eggs after crosses of irradiated females with irradiated males treated at the same dose. Error bars show standard error (n = 15 crosses per dose).

**Figure 4 insects-11-00564-f004:**
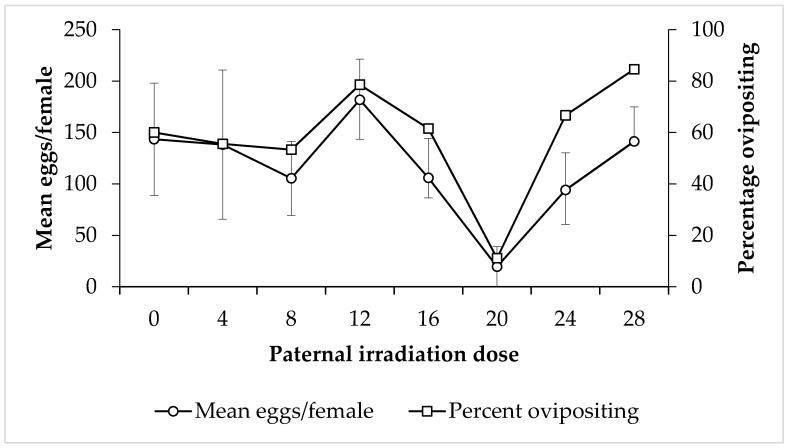
Mean number of eggs laid by untreated female *Nezara viridula* after mating with irradiated males (left axis), and the percentage of these females that laid eggs (right axis), separated by parental male irradiation dose. Error bars show standard error.

**Figure 5 insects-11-00564-f005:**
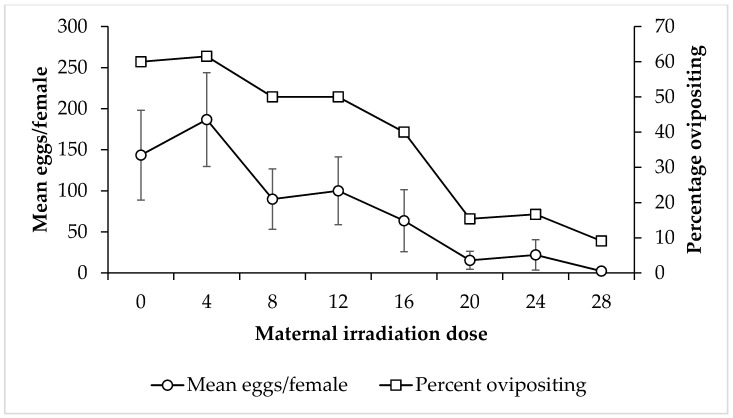
Mean number of eggs laid by irradiated female *Nezara viridula* after mating with untreated males (left axis), and the percentage of these females that laid eggs (right axis), separated by parental female irradiation dose. Error bars show standard error.

**Figure 6 insects-11-00564-f006:**
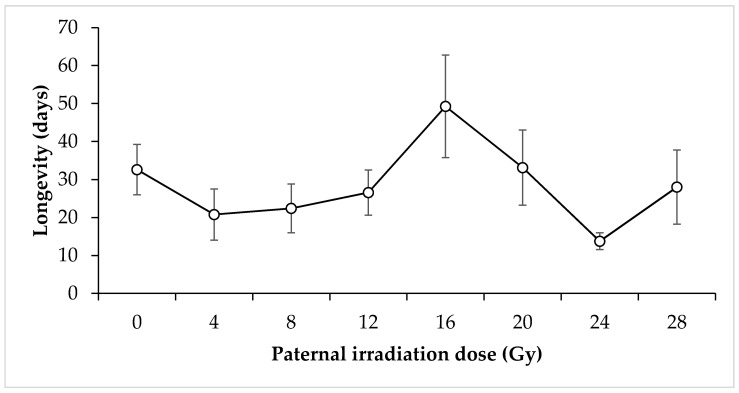
Mean longevity of irradiated male *Nezara viridula* that were crossed with untreated females. Error bars show standard error.

**Figure 7 insects-11-00564-f007:**
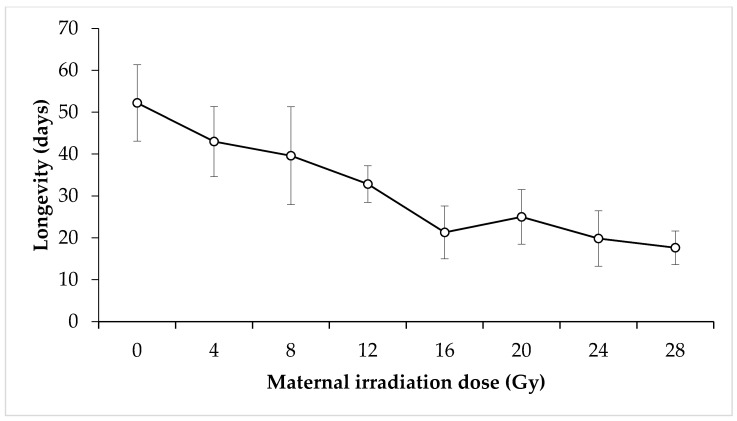
Mean longevity of irradiated female *Nezara viridula* that were crossed with untreated males. Error bars show standard error.

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
