# Peer review of "Egg Sterilisation of Irradiated Nezara viridula (Hemiptera: Pentatomidae)"

_insects, 2020, doi:10.3390/insects11090564_

Round 1

Reviewer 1 Report

The authors describe an interesting field of research for pest control. Target species (N. viridula) is a relevant pest worldwide and the discussion involve also a second severe pest, H. halys. SIT is an interesting technique in pest control, however, in the field of the target and the other cited species (above all H. halys), it is premature and at date, only lab experiences were performed.

In my opinion, the MS deals with and interesting aspects under the insect biological concepts, but authors should be more careful in the first part of the text with potential field use. I mean that the use of the concepts of eradication, mass-rearing, mass-sterilizing, and mass-release of N. viridula (or H. halys) is a demanding task, a hard challenge. This due to a series of factors that are in the first part ignored and lead the reader to intend a potential successful application in the field; reading abstract and introduction there is a high expectation of potential successful use in the field in the future. Only in the discussion, the authors describe many detailed aspects of critical factors for field application, all congruent and appropriate, which give to the reader the real perspective of potential use in the field, which is not so simple like it is induced to think when reading the introductive part. Thus, I would suggest being more realistic also in the introductive part.

Line 18: delete “of the opposite sex”. It is redundant.

Line 18: “mortality”: does not induce SIT sterility? i.e. females lay sterile eggs that not hatch. From a given point of view, this should not be considered as dead eggs.

Line 24: about future application: I'm skeptic of the use of SIT against pentatomids, and especially on N. viridula or H. halys. There are many different crucial factors that need to be considered before and are not irrelevant for practical field applications. 1) there is the need to rear juveniles up to adults and then sterilize them. Or it is foreseen the catch wild adults and release after sterilization? In the first option, it is severely time and resources consuming rear for more than one month juveniles. 2) I suppose that there is the need for thousands, or millions, of insects to rear, sterilize and release. 3) both cited pentatomid couple more times during their adult life and are long-living, thus the possibility to encounter a wild male is high. 4) BMSB and SGSB have a wide range dispersion ability, thus immigration is frequent and damage occurred anyway. 5) females are able to overwinter already fecunded. 6) adults, when molted, not couple immediately, but continue to feed. 7) there is a need for a high percentage of sterile specimens in the field. Have the author an idea of what should be the minimum percentage for success? I mean, in infested sites, especially of H. halys, there is an enormous population of adults and juveniles, millions. I suppose that 50% of sterilized specimens is not sufficient, maybe more than 80% is necessary, or more. Is this feasible practically? 8) when releasing thousands or millions of sterile insects, these insects feed for months on fruits and vegetables causing damage anyway. A Different scenario would be when adults are not feeding or not damaging cultures, but it is not the case here.

Line 33: “This family”, which one’? It is to intend Pentatomidae, but not clear that authors refer to pentatomidae. Pentatomidae is only cited among brackets in the previous sentence and not specifically.

Line 37: “The resulting damage”: Please add more details on damages, or remove this sentence. It does not display relevant information in this way.

Line 41-42: please add a more complete list of approaches or delete this part of the sentence; in the way as presented here, it is incomplete. Biological control of BMSB is maybe a valid perspective, but I'm skeptic of the fact that vibrations could be a practical solution, very interesting under the biological and ethological aspect, but about field application? At least I would stop the sentence with biological control. Moreover, in SGSB what are the experiences in biological control under field application?

Line 43: “Approved”: what is to intend with "approved"? Approved by who? Management guidelines? Or simply tested under lab studies?

Line 45: “Other”: other what? Please, be more detailed.

Line 45: “eradicadion”: I suppose this is an excessive promising perspective. The literature cited here, on what kind of insect refer? I suppose that today nobody can reasonably hope to eradicate BMSB or SGSB from an area, at least if not in the case of a small Island. At least it would be possible to mitigate negative effects with lowering the pest populations, but with which costs of production? And which proportion of success?

Line 47: “mass rear”: is it realistic mass-rear BSMB or SGSB? mass rearing means millions of specimens.

Line 47: “mate”: females couple only one time in their life? So far as I know, no.

Line 57: “H. halys”: so far as I know, the most interesting and realistic perspective to contorl the pest is through the use of Trissolcis japonicus. The fact that it is an exotic species with all its ecological consequences, is mitigated by the fact that several adventive populaiton were found in the Northamericas and Europe.

Line 57-60: move this part above, after line 42. It is about general investigation of control measures

Line 60: “extinction”: In SGSB and BMSB, this is possible only on an island (better if a small one) or in an other type of physically isolated area. Immigration is a continuous factor. This, hypothetically, can be possible for New Zealand for example, or minor islands, but not for countries in the Palearctic or Nearctic area.

Line 60: separate in a new paragraph

Line 64: “eradication”: in pentatomids, I think that it is a too high expectation to eradicate with SIT. Adults are long-living, for months and couple more times. Moreover, sterile insects continue to feed which produce a damage. In high-quality fruit production, any kind of damage is economically relevant. What happens if millions of adults even sterile are released in the field? Where are they going to feed? I suppose damage insurance is to be foreseen.

Line 66-67: “successful …..”: SIT in pentatomids is an interesting approach under the biological point of view, but I suppose not in pest management in the practice.

Line 74: “fifth instars”: does this mean that the intention for a practical application is to release in the field 5th instars? This means that it is to release an organism that feeds on fruits and vegetables for many weeks prior to couple and disrupt reproduction

Line79: “data”: which data? please cite literature or if unpublished data

Line 94: “larvae”: is this correct for a gradual metamorphosis insect?

Line 107: “egg mortality”: egg mortality or egg sterility? I suppose that the topic is egg sterility due to the fact that females coupled with sterile males, thus laid sterile eggs. Or am I wrong? please, can authors explain better? In pentatomids, unfertilized eggs are dead eggs or simply not hatch because not fertile?

Line 115: “similar”: similar to what? I suppose to the previous results but it is not clear cited.

Line 118-119: this part, is a discussion of the data.

Line 121: “sterility”: sterility of females or mortality of the eggs? please use in all the cases the same concept to avoid confusion. Here the graph is about egg mortality. the same in figure 1

Line 131, paragraph “Fertility”: the case of egg production in bi-parental treatment is missing

Line 133: this is a discussion

Line 141, 144, 157 …… : use plural for both male and female when in the same sentence. check through text in more parts.

Figure 6 and 7: move the x-axis left from circles and square, like done for example in figure 5

Line 166: “sterility”: please check accurately the terminology, if sterile females lay sterile eggs or dead eggs. I suppose sterile eggs and not dead eggs. A dead egg is an egg that could potentially hatch but the embryo is dead due to different causes. Unfertilized eggs re dead eggs?

Reviewer 2 Report

Title: Egg Sterilisation of Irradiated Nezara viridula (Hemiptera: Pentatomidae)

This study looked at sterilization of eggs through radiation exposure of adult male and females. The authors looked at various combinations of male, female, or male and female sterilisation at various doses of radiation exposure. This work makes an important contribution to the application of the sterile insect technique, furthering our understanding of the level of radiation needed to cause subsequent sterility in another important species.

The paper was quite concise with excellent dose response curves providing clear indication of the successful radiation dose required for sterility, and measures of longevity of adults after exposure.

I have some minor edits for the authors.

The last paragraph of the introduction is awkwardly written. It appears to be several different incomplete thoughts forced together. Please rework this paragraph.

In all figures have caption match Y-axis label. Choose either sterility or mortality and have these terms match.

In all figures remove the “%” after each number on the Y-axis, it is already given in the axis label.
